# Intrinsically Disordered Proteins: An Overview

**DOI:** 10.3390/ijms232214050

**Published:** 2022-11-14

**Authors:** Rakesh Trivedi, Hampapathalu Adimurthy Nagarajaram

**Affiliations:** 1Department of Translational Molecular Pathology, The University of Texas MD Anderson Cancer Center, Houston, TX 77054, USA; 2Laboratory of Computational Biology, Department of Systems and Computational Biology, School of Life Sciences, University of Hyderabad, Hyderabad 500046, Telangana, India

**Keywords:** protein structure, protein function, intrinsically disordered proteins, intrinsically disordered regions

## Abstract

Many proteins and protein segments cannot attain a single stable three-dimensional structure under physiological conditions; instead, they adopt multiple interconverting conformational states. Such intrinsically disordered proteins or protein segments are highly abundant across proteomes, and are involved in various effector functions. This review focuses on different aspects of disordered proteins and disordered protein regions, which form the basis of the so-called “Disorder–function paradigm” of proteins. Additionally, various experimental approaches and computational tools used for characterizing disordered regions in proteins are discussed. Finally, the role of disordered proteins in diseases and their utility as potential drug targets are explored.

## 1. Introduction

The functional aspects of genes have been attributed to RNAs and proteins. Of the two, proteins are the ones that bring about the majority of diverse cellular effector functions. Two paradigms concerning the structure and function of proteins have evolved, as shown in Figure 1 [1,2,3,4,5,6]. The first paradigm corresponds to the well-established, often assumed as the default for proteins, ‘structure–function paradigm’, which states that the three-dimensional native structure under physiological conditions is the prerequisite for a protein to function. The second paradigm is the recently established ‘disorder–function paradigm’ based on the proteins that perform cellular functions without attaining a stable three-dimensional structure under physiological conditions [7]. The naturally occurring, biologically active proteins that appear to possess a high degree of conformational flexibility have been referred to as intrinsically disordered proteins (IDPs) [5,8,9,10]. In most instances, instead of the whole protein, only some regions in the protein are disordered and functional; such protein segments are known as intrinsically disordered regions (IDRs) [11,12,13,14]. Interestingly, these intrinsically disordered proteins/regions have endowed proteins with functional promiscuity [15,16,17,18].

### 1.1. Structure–Function Paradigm

In the early stages, protein research was focused on studying globular (ordered) proteins; therefore, the principles laid to explain the protein structures were more specific to globular proteins. These fundamentals defining the protein structures received continuous experimental support throughout the 20th century from the multitude of protein structures submitted to repositories, such as that of the Protein Data Bank (PDB) [19]. There was no doubt about the relatedness of the protein structure and function. Hence, the central philosophy of structural biology was described as follows: “a protein requires a native folded structure to perform its biological function”.

Proteins were initially considered as rigid, crystal-like biological molecules with no flexibility. This “static” view of protein function was supported by one of the best-understood protein functions, catalysis [20]. This static view of proteins led to the assumption that the complementarity of shapes at the binding interface and the emerging non-covalent forces between molecules are the major drivers of interactions involving proteins [21]. After Koshland proposed the “induced-fit model” of catalysis, the theory of presence of some degree of flexibility in protein structures to carry out their biological functions was accepted [22].

The three-dimensional native structure of a protein, which is at the center of the structure–function paradigm, is encrypted in the protein’s amino acid sequence and is governed by the intricate balance of various physical forces between its atoms [23]. Physical forces include strong covalent (peptide/amide) bonds, which connect atoms of the polypeptide sequence, and weaker attractive/repulsive forces between non-bonded atoms. All of these fundamental interactions act coherently and assist a polypeptide in attaining a well-defined three-dimensional structure under physiological conditions. A unique 3-D structure of a protein ensures a precise spatial arrangement of residues, which provides a particular physicochemical microenvironment required for the extremely specific catalysis of chemical reactions, binding of ligands, ion or small molecule translocation, or macromolecular complex assembly [24]. Along with the deposition of a large number of protein structures into the PDB, the availability of the mechanistic details of a protein’s functions, such as enzyme catalysis, receptors, transporters, membrane channels, etc., also helped the acceptance of the structure–function paradigm. However, with increasing reports of the structural/functional data deviating from the classical structure–function paradigm, there was a need to reassess this well-established paradigm, which, in turn, led to the proposition of the disorder–function paradigm.

### 1.2. Disorder–Function Paradigm

The early recognition of the significance of protein structures in their functioning is generally considered the prime reason for the prolonged negligence of protein disorder. Unusual behavior of proteins, such as missing electron density in the PDB structures, increased sensitivity to in vitro proteolysis, peculiar behavior during the purification process, etc., has always been observed, but in the light of the classical structure–function paradigm, these observations were initially considered as artifacts. This “dark side” of structural biology underwent an intensive reassessment against the backdrop of proteins’ considerable inherent flexibility and peculiar properties, and resulted in the recognition of the intrinsic structural disorder of proteins as a standard feature across proteomes. Altogether, the widespread occurrence of intrinsically disordered proteins or protein segments and their functional relevance has led to the introduction of a new generalization with respect to the protein structure and function, i.e., the ‘disorder–function paradigm’.

Intrinsically disordered proteins (IDPs) or regions (IDRs) remain unfolded and carry out their biological role under physiological conditions [25]. Just as in the case of ordered/globular proteins, the structural organizations of IDPs/IDRs are also governed by the same physical forces, albeit with some differences in their balance and emerging protein structure dynamics [26]. Most of the early concepts regarding the structure of IDPs/IDRs came from the unfolding/folding reaction studies of globular proteins. At the level of genome organization, large proportions of gene sequences have also been observed to code for functional long amino acid stretches, which most likely either attain a non-globular conformation or exist as unfolded entities in solution as IDPs/IDRs [27].

Over the years, our understanding of the protein structure and function has increased tremendously. However, with the increasing reports of the involvement of protein disorder in carrying out cellular functions and the development of computational tools to predict disordered residues of proteins, there has been an exponential increase in interest in studying IDPs/IDRs. This review article summarizes various features of intrinsically disordered proteins/regions that form the basis of this newly established ‘disorder–function paradigm’.

## 2. Intrinsic Protein Disorder

Repeated occurrences of proteins with intrinsic flexibility and properties different from those of ordered/globular proteins gradually resulted in the development of the notion that non-rigid proteins are not exceptions. Around 2000, these naturally flexible proteins were accepted as a general category of proteins [27,28,29,30]. The conformational flexibility of these “non-traditional” proteins was proposed to be the source of their biological functions [31]. Over the years, various terms have been introduced by different authors to describe these proteins with inherent flexibility. It was only in recent years that the phrase “intrinsically disordered proteins” (IDPs) became more widely used than other terms [28]. Most acceptably, “intrinsic protein disorder” defines the biologically active proteins or protein segments that exist as ensembles of unfolded, collapsed, extended, non-globular conformations at the secondary or tertiary structural level [27,28,29,30,32,33].

### 2.1. Natural Abundance of Intrinsically Disordered Proteins

Information about IDPs was very sparse for a very long time. Until the reports on the experimentally characterized IDPs, it appeared improbable to have such a class of proteins in abundance [34]. However, computational predictions revealed the possible widespread occurrence of IDPs and IDRs [35,36,37]. The exhaustive analysis of 31 genomes spanning three kingdoms of life revealed that a considerable number of proteins contain regions with 40 or more consecutive disordered residues [28]. The proportion of structural disorder was also found to increase progressively with genome complexity, from bacteria to archaea, and then to eukaryotes [28,38]. While 33% of eukaryotic proteins have been reported to contain at least one functionally relevant long (>30 residues) intrinsically disordered region, archaean and eubacterial proteins possess only 2.0% and 4.2% of such functional IDRs, respectively [36]. In viral proteomes, the total intrinsic disorder content is determined by the nature of the nucleic acid constituting the viral genome, and it decreases successively with an increase in the size of the viral proteome [39]. Recently, 3133 unique proteins were experimentally validated to contain functional long disordered regions (at least 30 residues) [40]. Additionally, the degree of disorderedness in proteomes and essential proteins was estimated for various genomes, and a sharp increase was observed at the prokaryote/eukaryote boundary [41]. Thus, the natural abundance of disordered proteins or protein segments across complex genomes suggests that, even though IDPs/IDRs fail to attain stable three-dimensional structures under physiological conditions, they are of high functional pertinence [27,28,29,30,42,43,44,45,46,47].

### 2.2. Sequence Characteristics of Intrinsically Disordered Proteins

The propensity of a protein or protein segment to fold or remain unfolded under physiological conditions is encrypted in its amino acid sequence [28,30,32,33,45,46]. In other words, the amino acid sequence composition determines whether a protein would be an ordered protein with a stable folded 3-D structure or an unfolded intrinsically disordered protein. Strong electrostatic repulsions due to a higher net charge and a lack of driving force for compaction due to low mean hydrophobicity are generally considered as the prime reasons for the unfolded, extended structure of IDPs/IDRs [45].

An in-depth comparative analysis of the sequence composition of ordered and disordered proteins revealed that residues such as Ala, Arg, Gly, Gln, Glu, Lys, Pro, and Ser (referred to as disorder-promoting residues) occurred more frequently in IDPs/IDRs. In contrast, residues such as Asn, Cys, Ile, Leu, Phe, Val, Trp, and Tyr were more common in the ordered/structured segments of the proteins (referred to as order-promoting residues) [28,48,49,50]. Comparative studies of amino acid residues in disordered and ordered regions, physicochemical property-based scales (such as the coordination number, aromaticity, strand propensities, flexibility index, volume, helix propensities, etc.), and composition-based features (e.g., any combination that has one to four residues in the group) have led to the distinction between disordered and ordered regions in proteins [28,50]. Compositional bias and sequence characteristics play a significant role in defining the interactions of disordered proteins/regions [51,52]. Furthermore, the distinct compositional bias of the intrinsically disordered proteins/regions as compared with the ordered proteins/regions forms the basis for developing many disorder-predicting computational tools. For instance, tools involving the alignment of IDPs/IDRs use specific amino acid substitution scoring matrices reflecting the frequency of occurrence of different residues in the disordered regions of proteins [53,54,55,56].

### 2.3. Structural Aspects of Intrinsically Disordered Proteins

In general, most proteins exist as a combination of both ordered and disordered segments in different proportions [57,58,59,60,61,62]. Unlike the ordered regions of proteins, which exist as stable secondary/tertiary structures, the disordered regions fail to attain a stable three-dimensional native structure under physiological conditions. However, the structure of IDPs/IDRs can be best defined as the ensemble of functionally relevant interconverting transient structures on a fast time scale.

Several previous studies reported that IDPs/IDRs are enriched with uncharged and polar amino acids, lacks bulky hydrophobic residues, and exist as dynamic heterogeneous ensembles of collapsed or extended structures [63,64,65,66]. Furthermore, IDPs/IDRs exhibit different degrees of foldability, ranging from potentially foldable to not foldable at all [59,60,61,67].

IDPs/IDRs do not possess a precise equilibrium value of the atomic coordinates and backbone Ramachandran angles over time; as a result, they appear as “protein clouds” [68]. Despite being highly dynamic, the structures of such protein clouds are best described as a few low-energy conformations [69,70].

In most cases, upon interaction with specific binding partners, a disordered protein/segment undergoes a disorder to order conformational transition (termed as ‘induced folding’) [27,28,42,43,44,47,70,71,72,73,74,75],. Additionally, a disordered protein can bind to multiple partners to attain distinct conformations with each of them, which, in turn, enable it to interact with different targets [28,76,77]. Moreover, there is also a “conformational preference” for the structure attained by IDPs upon binding [78,79]. Post-translational modifications (PTMs) also enable IDPs/IDRs to attain diverse conformations, thus increasing the total repertoire of structures resulting from an IDP/IDR sequence [42,72,80,81,82,83,84]. Ensembles of the conformations resulting from a single sequence make it possible for IDPs to perform multiple apparently unrelated biological functions (termed as ‘moonlighting’) required for the maintenance of life [85,86,87,88,89].

### 2.4. Functional Classification of Intrinsically Disordered Proteins

Several classification schemes have been proposed over the years based on the functions performed by IDPs/IDRs [42,90]. Tompa et al. annotated IDPs/IDRs in six different functional categories depending on the presence/absence and the strength of the binding of the disordered proteins/regions to their ligands [29,91]. Later, this stratification was further extended to define eight functional classes of IDPs, namely entropic chains, modification sites, disordered chaperones, molecular effectors, molecular recognition assemblers, molecular recognition scavengers, metal sponges, and unknown, as shown in Figure 2 [92]. These functional subtypes can be present either alone or in combination within the same protein if the protein has several disordered regions [59,60,61]. In the following sections, different functional classes are described in some detail with relevant examples.

#### 2.4.1. Entropy Chains

Entropic chains take advantage of their conformational flexibility and perform biological functions without becoming a structured entity [93,94]. Examples of entropic chains include flexible linkers and spacers [29,91]. Flexible linkers regulate the relative movement of domains positioned at the two ends of the linkers, whereas spacers specify the inter-domain distances. In addition to providing a means of distance regulation between domains/regions within proteins, both flexible linkers and spacers enable extraordinary freedom for inter-domain orientations. In general, the conformational ensembles generated through computational simulations and modeling are used to understand the flexibility and positioning of the domains in multidomain proteins [95].

Several previous studies showed the significance of conformational flexibility for IDPs/IDRs to function. For instance, a mutation in the flexible linker region connecting light and heavy chains of multidomain antibody protein reduces the linker’s flexibility, disturbs the domain’s orientation, and limits their interactions [96]. In another study, disordered and flexible linker regions of the host proteins have been shown to be the primary targets of the oncogenic adenoviral Early region 1A (E1A) protein [97]. Replication protein A subunit (70 kDa) manifests a conserved dynamic behavior regardless of the insignificant sequence conservation, which is also evidence of the importance of flexibility from the perspective of functional relevance [98]. The microtubule-associated protein 2 (MAP2) projection domain facilitates the spacing of cytoskeletal structures by repelling molecules that approach microtubules [99]. The disordered regions in the mammalian CIP/KIP family of proteins act as flexible linkers or spacers, and have a conserved propensity to remain in the disordered state [100]. Along with the kinetic properties of the binding partners, the linker flexibility and length are critical factors in determining the processivity of enzymes [101].

#### 2.4.2. Modification Sites

Most of the regulatory and signaling proteins possess IDRs, and the intrinsic flexibility of these regions as display sites affords them an advantage over ordered regions [10,13,36,59,90]. Furthermore, the prevalence of charged side chains and low-complexity regions in the disordered proteins facilitates their multivalent electrostatic interactions with the charged lipid head groups in the membranes. Thus, both the composition of membranes and the disorderedness of IDPs/IDRs fine-tune the signaling pathways [102]. In several independent studies, approximately 15% of all disordered proteins have been reported as lipid-binding proteins [103,104].

The flexibility of IDRs helps them to undergo transient, but specific, interactions with enzymes carrying out various post-translational modifications [105,106,107]. Moreover, this intrinsic flexibility of IDRs facilitates the easy entry and recognition of post-translational modifications within IDRs by effector molecules, leading to subsequent outcomes, such as protein stability, turnover, and localization within a cell upon binding [102,105,108]. p53, p27, histone protein tails, and the CREB-kinase-inducible domain are a few examples of well-explored IDPs where PTMs are critical for functioning and regulation [108,109,110]. Recently, phosphorylation at multiple sites within the disordered regions of four representative IDPs (Ash1, E-Cadherin, CTD2, and p130Cas) has been reported to regulate the IDR-mediated protein–protein interactions through conformational plasticity [111]. Furthermore, site-specific phosphorylation within the intrinsically disordered AF1 domain of glucocorticoids receptor (GR) generates surfaces appropriate for the interaction of the AF1 domain with coregulatory binding proteins (BP), which, in turn, regulates the transcription of the downstream genes involved in GR signaling [112]. Additionally, the methylation of arginine residues in low-complexity and disordered regions has been shown to regulate genome integrity, gene transcription, splicing, mRNA–protein complex biology, protein translation and stability, and phase separation [113].

#### 2.4.3. Disordered Chaperones

Proteins that help RNAs and other protein molecules to attain their functional folded state are known as chaperones [114,115]. About half of the RNA chaperone sequences and one-third of the protein chaperones are disordered in nature. The ability of IDRs to adapt their structures according to different binding partners and increase the lifetime of the encounter complex during binding supports their chaperone activity [116,117]. Disordered chaperones prevent toxic aggregation by quickly binding and solubilizing misfolded proteins. In addition, the transient binding of unfolded or misfolded substrate molecules to disordered chaperone regions helps them to fold/refold correctly in a thermodynamically permissible manner [118]. hnRNP A1, GroEL, α-crystallin, Hsp33, etc. are a few well-known examples of IDPs with chaperone activity [119,120].

RNA chaperones enriched with positively charged residues promote the compactness of nucleic acids conformation by acting as a counterion that locally shields repulsive electrostatic repulsions [121]. For instance, the dimerization of the hepatitis C virus genome is facilitated by its core protein with chaperone activity [121]. Additionally, disorder-enriched RNA binding proteins (RBP) have been found to stabilize the disorderedness of the interacting IDPs/IDRs and maintain their functional competence, thus making the disordered state of the human proteome available for drug targeting [122]. Furthermore, IDRs present in the human chaperones Hsp110s, Hsp105α, and Apg-1 prevent general protein aggregation and amyloid genesis [123]. With aging, the disordered chaperone-mediated quality control of intrinsically disordered proteome is lost, leading to the accumulation of protein aggregates responsible for neurodegenerative diseases [124]. In plants, dehydrins, the largest group of disordered chaperones, are expressed at higher levels under various abiotic stress conditions, such as drought, osmotic stress, or high temperatures. An intricate interplay of ordered and disordered segments of these proteins is required for proper cellular protection [125]. Additionally, disordered chaperones have been reported to lose flexibility upon binding to a substrate, which helps them to act as a molecular on/off switch, as characterized in Hsp33 [126,127,128].

#### 2.4.4. Molecular Recognition Effectors

Effector functional class disordered regions bind permanently to other proteins and modify their actions. Such IDRs often undergo disorder to order conformational transition upon binding to their interacting partners, which is known as ‘coupled folding and binding’ [129,130,131,132,133]. The degree of folding upon the development of intermolecular contacts may vary significantly. The disordered protein partners may either fold completely or remain as an extended structure with short flanking ordered regions [134]. Recently, a ubiquitin-binding domain (referred to as DisUBM) was identified in many proteins that remained mostly disordered in a free state, and attained an α-helical structure upon binding to ubiquitin. Such naturally occurring DisUBMs are considered as general affinity enhancers of IDPs that can bind to fold proteins with a possibility of ubiquitinylation [135]. In general, disordered regions have a kinetic advantage over folded segments, even after intermolecular interactions are formed during the folding-upon-binding process [136].

Effector-disordered regions can also alter the functioning of the other parts within the same protein in two different ways: (i) competitive interactions and (ii) allosteric modulation. For example, the disordered GTPase binding domain (GBD) of Wiskott–Aldrich Syndrome protein (WASP) controls its autoinhibition through competitive binding [137]. Upon binding to cdc42 protein there is an enhanced interaction of the WASP GDB domain with the actin cytoskeleton regulatory machinery. However, GDB interaction with the actin machinery is inhibited as it folds back on the other region of WASP. In another study, IDRs of general transcription factors and transcriptional regulators were shown to compete for the folded TAZ1 domain of CREB-binding protein (CBP) in response to various types of stress stimuli. For instance, under oxygen-deficient conditions, the hypoxia-inducible transcription factor (HIF-1α) C-terminal transactivation domain drives the transcription of different critical adaptive genes. One such gene translates to HIF-1α negative feedback regulator protein CITED2, which represses the transcriptional activity of HIF-1α by directly competing for CBP’s TAZ1 binding site [138]. Allosteric coupling in disordered regions can be represented by the binding of adenovirus E1A oncoprotein with retinoblastoma protein (pRb) and the TAZ2 domain of CBP [139]. The binding of pRb to E1A increases the probability of CBP binding, whereas the initial binding of CBP to E1A decreases the possibility of pRb binding. In addition, the globular KIX domain of the CBP is known to bind cooperatively with CREB kinase-inducible transactivation domain (pKID), the activation domain of transcription factor c-Myb, and mixed-lineage leukemia (MLL) protein. The binding of the pKID and c-Myb to the KIX domain is mutually exclusive and regulated by the allosteric binding of MLL [140,141,142]. These results also suggest that the allosteric coupling can also be determined by the stability of the individual protein conformations achieved when interacting partners are binding [143,144].

#### 2.4.5. Molecular Assemblers

As the name suggests, assemblers interact with multiple binding partners to encourage the formation of higher-order protein complexes, such as the ribosome, activated T-cell receptor complexes, and transcription pre-initiation complexes [91,145,146]. An assembly of various interacting partners of large complexes is feasible due to the multiple types of disordered functional regions, such as short linear peptide motifs (SLiMs), molecular recognition features (MoRFs), etc. In fact, with an increase in the size of the protein complexes, the disorder content of the assembler protein increases [147]. Furthermore, upon binding to their partner proteins, disordered assemblers maintain their open structure, allowing multiple proteins to bind to a single IDR [148,149]. Therefore, the overall architecture of assemblies formed by IDPs depends on the disorder content, the available number of interaction sites, and the nature of the binding molecules [150].

The assembler function of IDPs is an outcome of their multivalency and can be achieved in two distinct ways. Firstly, IDPs can act as the binder molecule by tethering all of the interacting proteins and stabilizing the complex. For example, ribosome assembly involves the cooperative binding of proteins enriched with IDRs and RNA [151]. Secondly, IDPs can also regulate the spatiotemporal assembly of the different signaling partners by acting as a scaffold, such as the IDP-mediated formation of biomolecular condensates, nuclear pore complex, and cytoskeletal assemblies [2,152,153]. Among all the functional categories, the scaffolding regions of the assemblers contain the highest degree of disorder [149].

#### 2.4.6. Molecular Recognition Scavengers

This functional class of intrinsically disordered proteins/regions stores and neutralizes small ligands. For example, caseins and other calcium-binding phosphoproteins (SCPPs) are highly disordered proteins that can act as scavengers by solubilizing calcium phosphate clusters in milk and other biofluids [154]. The proline-rich salivary gland glycoproteins also belong to the molecular recognition scavengers class of IDPs; they function by binding and neutralizing the tannin molecules in digestive tracts [29]. Plant dehydrins are highly disordered proteins expressed at elevated levels in the late stages of embryogenesis. The VviDHN4 isoform of dehydrin acts as a scavenger by removing reactive oxygen species from the cellular environment [155]. Human peroxiredoxins-4 protein, most highly expressed and localized exclusively within the endoplasmic reticulum, acts as its major hydrogen peroxide scavenger [156]. Other examples of IDPs with scavenger activity include chromogranin protein A, which is responsible for sequestering adrenaline and ATP in the medulla of the adrenal gland [154].

#### 2.4.7. Metal Sponges

IDPs/IDRs that are capable of storing and neutralizing heavy metals belong to the metal sponges category, such as the sulfonation of catecholamines in humans by catecholamine sulfotransferase enzyme (Sulfotransferase 1A3). This reaction is a detoxification pathway because it readily forms excretable water-soluble metabolites [157]. Additionally, the SmbP protein of the ammonia-oxidizing bacterium *Nitrosomonas Europaea,* binds to divalent cations, especially copper, to prevent cellular toxicity [158]. Similarly, the crystallin proteins from the bacteria *Yersinia Pestis* act as metal sponges by managing the cellular copper levels [159].

All of the other IDPs/IDRs with no experimental support or evidence proving/disproving their role in protein function were assigned to the unknown class. Upon functional annotation, the members of this class are transferred to one of the above-mentioned functional categories.

### 2.5. Functional Elements of Intrinsically Disordered Proteins

Various functional regions in intrinsically disordered regions have been revealed when studying different classes of functions carried out by IDRs. In general, the functional modules within IDRs have been classified into three categories: (i) Short Linear Motifs (SLiMs), (ii) Molecular Recognition Features (MoRFs), and (iii) Intrinsically Disordered Domains (IDDs) [78,160,161,162,163,164,165]. The classification and features of each of these functional modules are shown in Figure 2. The following sections discuss these functional modules of disordered proteins/regions interactions.

#### 2.5.1. Short Linear Motifs

Short Linear Motifs (SLiMs), also known as MiniMotifs or Linear Motifs (LMs), are 3–10-residue-long peptide segments that occur within IDPs [160,166,167]. SLiMs typically comprise a stretch of loosely conserved amino acids interspersed with a few highly conserved residues [160]. As a consequence, interactions mediated by an individual SLiM motif have weak affinities. However, multiple SLiMs frequently coordinate to produce higher-affinity dynamic interfaces. Even though SLiMs are short sequences with transient binding capabilities, they are indispensable to a protein’s binding precision and functioning. The Eukaryotic Linear Motif (ELM) initiative aims to characterize and annotate motifs, mostly focusing on, but not limited to, eukaryote proteins. At present, 1,000,000 SLiMs have been predicted across the human proteome, but only a small fraction of it has been explored in detail to date [166,168,169,170,171].

Owing to their crucial role in regulating signal transduction and protein–protein interactions, SLiMs are considered as functional in almost every biological pathway. They can interact with a variety of biological entities, including globular protein domains, intrinsically disordered proteins or regions, RNA, lipids, etc. Broadly, SLiMs can be classified into two distinct groups: modification sites and ligands, with each having multiple subgroups [2]. If a linear motif act as a modification site, catalytic domains of enzymes interact with SLiMs to catalyze cellular processes, such as post-translational processing, the addition or removal of moieties, and structural alteration of the peptide backbone [172,173]. In contrast, when linear motifs behave as ligands, they act as complex-promoting motifs required for scaffolding and increasing the avidity of interactions with binding partners. Additionally, SLiMs can act as a docking motif to increase the specificity and efficiency of modification events, and as a targeting/trafficking motif to provide stability to proteins and facilitate their subcellular localization and trafficking [174,175,176,177,178,179]. For example, in the family of Rho-activated kinases (Raks), kinase function activation and uniform subcellular localization is an outcome of cleavage by caspases at SLiMs identified in disordered regions [180]. The binding of the catalytic and regulatory subunits of protein serine/threonine phosphatases (PPP) and their substrates depends on SLiMs [181]. The interaction of human-origin recognition complex subunit ORC1 and licensing factor CDC6 during pre-initiation complex formation and its regulation by CDKs are also mediated by SLiMs [182,183].

Furthermore, SLiMs can also act as molecular switches by allowing the mutually exclusive binding of different partners at the protein’s same or distinct SLiM sites [184,185,186]. Such SLiM features are essential to bring about the context-dependent moonlighting functions of IDPs.

#### 2.5.2. Molecular Recognition Features

Molecular Recognition Features (MoRFs) or Elements (MoREs) are 10–70-residue-long peptide motifs that can occur in the disordered regions of proteins [163,164]. These motifs undergo disorder-to-order transition upon binding with their partners (i.e., folding upon binding) and attain well-defined structures, such as α-helices (α-MoRFs), β-strands (β-MoRFs), γ-coils (γ-MoRFs), or mixtures of all of these conformations (complex-MoRFs) [162,163,187]. The final conformation achieved after transition may occur either due to the conformational preference of unbound MoRFs or the induced folding after binding [78,130,188]. The structural preorganization of MoRFs is modulated mainly by the sequence itself [189]. Therefore, sequence analysis of MoRFs from the structural, conformational, and interaction mechanism aspects will help to understand the properties of MoRFs [190]. A recent detailed computational investigation of 868 complete proteomes showed that 29% of IDRs of bacteria and archaea and 21% of eukaryotes harbor MoRFs [191].

p53 protein is a classic example of a protein with multiple MoRFs in a disordered state attaining multiple conformations upon interaction with different binding partners [163,164,192,193]. MoRF at the N-terminus of p53 protein is the primary binding site of Mdm2 and the other 40 proteins. The MoRF between residues 40 and 60 is the binding site of the Mdm2 and RPA70 proteins. C-terminal MoRF binds to multiple proteins, including S100B and the cdk2–cyclin-A complex. Each of these unique combinations of MoRF and the interacting proteins results in a distinct conformation of p53.

Moreover, the T-cell surface glycoprotein CD3 zeta chain has two MoRFs, referred to here as N-terminus and C-terminus MoRF. While tyrosine kinase, tyrosine phosphatase, and nef protein bind to N-terminal MoRF, the SH2 domain of the SHC protein binds to the C-terminus MoRF [194]. Additionally, the ethylene response factor in plants has several identified MoRFs, and the binding of different interacting proteins to these motifs helps the plants to overcome multiple stresses and adapt to different biological niches [195]. In conclusion, a single MoRF can bind structurally diverse sets of protein partners to give rise to multiple distinct conformations.

#### 2.5.3. Intrinsically Disordered Domains

Sequence-based approaches have identified various protein domains that are entirely or largely disordered both in isolation and solution. They are mostly involved in the DNA, RNA, and protein binding (e.g., Wiskott–Aldrich syndrome protein (WASP)-homology domain 2 (WH2) of actin-binding proteins) [196,197,198]. In general, the intrinsically disordered domains are a long region of disorder (>20 residues) with conserved function, conserved sequence, and conserved disorder [196,197,198,199]. Moreover, at the genomic level, the protein domains acquired during evolution as an outcome of the extension of the existing exons or exonization of previously non-coding regions tend to be highly disordered [200]. Therefore, the exonization of the previously non-coding regions could be one way of incorporating disordered segments into proteins. Some well-known examples of disordered domains include cell cycle regulatory proteins p21, p27, p57, etc. The disordered domains of these proteins have been experimentally and computationally verified [201,202].

Interestingly, specific disordered regions have been observed to co-occur frequently with particular types of structured domains in the same sequence [203,204]. Some globular domains require the presence of short, disordered regions in their vicinity, while others require disordered segments at specific locations with respect to the domain boundaries [203]. These findings suggest that the functional module in proteins can comprise either disordered/ordered regions alone or in combination. For example, ER degradation-enhancing alpha-mannosidase-like proteins (EDEM) recognize misfolded proteins and direct them to the ER-associated protein degradation (ERAD) process. EDEM3 degrades the misfolded proteins with the help of both structured domains (mannosidase homology domain (GH47), intermediate (IMD), and protease-associated (PA)) and intrinsically disordered domains [205]. Furthermore, the Bcl-2 family mediates apoptosis, and the maintenance of stem-like behavior of glioblastoma multiforme spheres by RBM14 protein also requires both structured and disordered domains [206,207]. Therefore, studying the co-occurrence of structured domains and IDRs in the same protein may prove insightful in understanding the unannotated disordered segments of proteins with respect to their biological role.

## 3. Experimental Approaches for Assessing Intrinsic Protein Disorder

Intrinsic protein disorder can be recognized and characterized by various direct and indirect (bio)physical methods. In contrast to direct techniques, which provide structural information about proteins, indirect approaches do not offer any structural details. Still, they suggest a behavior from which the disordered nature of the proteins can be inferred.

### 3.1. Indirect Methods

Early understanding of the intrinsic structural disorders of proteins was based on a few simple techniques. In general, these indirect intrinsic disorder-identification approaches can quickly provide ample insight into the structural states of a protein or its segments.

Because of the unusual amino acid composition and lack of a compact hydrophobic core, disordered proteins are evident during the purification process. Usually, the molecular mass (M_w_) of IDPs estimated by sodium dodecyl sulfate–polyacrylamide gel electrophoresis (SDS–PAGE) is higher by a factor of 1.2–1.8 in comparison with that measured by mass spectrometry [29]. Indeed, due to the enrichment of acidic residues and extension in solution, IDPs bind less to SDS and migrate more slowly on the gel in comparison with globular proteins [208]. The aberrant mobility of IDPs is also observed in size-exclusion chromatography (SEC) or gel-filtration (GF) experiments, as a result of which the apparent M_w_ of proteins with disordered regions is higher [45]. Furthermore, the flexible regions of proteins are known to have increased sensitivity to proteolytic degradation. IDPs, which are more affected than ordered proteins, based on limited in vitro proteolysis, exhibit high inherent flexibility [29,42,45,72].

Other peculiar biochemical behaviors of IDPs/IDRs include insensitivity to high temperatures and stability under acidic treatment. The resistance of IDPs/IDRs to boiling temperatures and acidic pH values has been ascribed to their lower contents of hydrophobic residues and enrichment of polar/charged residues, respectively [209,210,211]. Neutralizing acidic groups at lower pH levels reduces the net charge on IDPs/IDRs, leading to their increased solubility and a more compact structural state [45]. In contrast to IDPs/IDRs, the aggregation/precipitation of globular/ordered proteins occur at elevated temperatures and under low-pH conditions. While high-temperature conditions expose the hydrophobic core of ordered proteins, acidic conditions cause protonation of their negatively charged side chains, leading to charge imbalances, followed by the disruption of salt bridges and aggregate formation [29].

### 3.2. Direct Methods

Several techniques provide both steady-state and dynamic structural information on IDPs/IDRs at the residue level. These methods capitalize on the significantly distinct conformational behavior of IDPs compared with that of globular proteins [29]. Some of the most commonly used direct methods are as follows:

#### 3.2.1. X-ray Crystallography

The diffraction intensity and X-ray pattern scattered by electrons in the protein structure are used to construct a three-dimensional (3-D) model of electron density, which, in turn, is used to deduce the atomic nuclei positions in the protein molecule [29]. Disordered regions in X-ray structures appear as missing regions [28]. This method can provide protein structure resolution down to 1Å. Still, additional experimental support is required to be certain about the structural disorder, as missing electron density regions can also result from technical failures in crystallography [212].

#### 3.2.2. Circular Dichroism (CD)

Circular dichroism (CD) is an absorption spectroscopy-based approach that relies on measuring the difference in the absorption spectra of right-handed and left-handed circularly polarized light. Optically active chiral molecules preferentially absorb either right-handed or left-handed circularly polarized light. Near-UV (250–350 nm) and far-UV (190–230 nm) CD signals are generally used to determine different aspects of the structure of proteins in solutions. The near-UV CD spectrum represents the tertiary structure around aromatic residues Phe, Tyr, and Trp [213,214]. While intense and detailed spectra characterize ordered proteins, those of IDPs are of low intensity and low complexity. The far-UV CD spectra of the secondary structural elements of proteins are quite distinct; therefore, they are used to determine the proportion of ⍺-helix, β-sheet, turn, PPII helix, and coil conformations in proteins [215]. If the far-UV CD spectrum is indicative of predominantly coil conformations, it indicates the disordered nature of the protein. In the case of proteins having both disordered and ordered regions, the CD does not provide clear information, as it lacks residue-specific details [28].

#### 3.2.3. Nuclear Magnetic Resonance (NMR)

NMR is the most common quantitative technique used for studying IDPs. The spinning ability of the charged atomic nuclei forms the basis of the 3-D structure determination of proteins in solutions using NMR. The directions of these spins are random, but the application of the external magnetic field can align these nuclei in directions either parallel or antiparallel to the applied magnetic field. These two states of nuclei have different energy levels, a low-energy state and a high-energy state. The low-energy state attains a high-energy state upon irradiation with electromagnetic radiation, and free inductive decay (FID) is obtained as the nuclei undergo relaxation. Fourier transformation of FID results in a NMR spectrum with peaks from different types of nuclei in the molecule, which, in turn, is used to characterize the local covalent and spatial arrangement of atoms [29,47,216,217].

In a protein, each nucleus of the individual residues experiences a different magnetic field depending on its microenvironment (referred to as the ‘shielding effect’ or ‘chemical shift’). The chemical shift of the peptide backbone (^1^H^⍺^, ^13^CO, ^13^C^⍺^, and ^13^C^β^) can be used to determine the secondary structure type of the given peptide segment [218,219]. Amino acids in ordered proteins are packed in different kinds of chemical environments, as a result of which their NMR spectrum resembles a combination of spectra of various secondary structure elements. In contrast, the NMR spectra of disordered proteins with extensive conformational averaging appear as a summation of the random coil spectra of residues of proteins [216,218]. In addition to the fine structural details, NMR also provides specific information at the residue level [28].

#### 3.2.4. Small-Angle X-ray Scattering (SAXS)

The SAXS technique can quickly define the structural characteristics of proteins of sizes ranging from a few kilo-Daltons to several giga-Daltons under various experimental setups [220,221,222,223]. Briefly, this method involves exposing samples placed in quartz capillary tubes to a collimated monochromatic X-ray beam source and capturing scattered photons with a detector [224]. Comparative analysis of the electron density distributions of the protein sample and pure solvent/buffer is then conducted to determine various parameters of the proteins in the solution, such as the molecular mass, volume, radius of gyration, folding state, etc. [220]. Moreover, SAXS data can also be used to define protein flexibility and the intrinsically disordered state of proteins in solutions [223,225]. The scattering profiles of the proteins obtained from SAXS experiments are most commonly represented as Kratky plots (s^2^I(s) as a function of s, where s and I represent the momentum transfer function and scattering intensity), which are used to obtain structural insights into the protein.

In contrast to globular proteins’ bell-shaped Kratky’s plot with well-defined maxima, disordered protein-specific Kratky plots exhibit a plateau for a given range of the momentum transfer function (s), followed by a monotonic increase [226,227]. Additionally, the experimentally determined radius of gyration (R_g_) of IDPs from the SAXS curve can be directly compared with the theoretical or experimental R_g_ values of a globular and random coil for a given number of residues. The R_g_ values of IDPs lie between those of highly compact globular proteins (lowest R_g_ values) and completely disordered/unfolded proteins represented by random coils (highest R_g_ values) [228]. Altogether, this method offers fast structural characterization of proteins in solutions with a relatively easy sample preparation protocol and can capture data under near-native conditions [229,230]. As the sensitivity of SAXS depends on the particle size, prior removal of the macromolecular aggregates during sample preparation using a method such as sedimentation or size-exclusion chromatography is suggested [231].

Finally, it is worth mentioning that SAXS-based studies of IDPs can use valuable a priori complementary information from several other experimental and in silico protein structure determination methods. For instance, X-ray crystallography depicts the structured regions of a protein, while SAXS defines the protein segments with missing electron density [232]. Similarly, NMR provides information about different domains/complex sub-units during analyses of bimolecular complexes and multi-domain proteins, and SAXS defines their relative inter-domain positions [233]. Furthermore, other complementary techniques, such as CD, spectroscopy, chromatography, etc., and SAXS, can be used for the biophysical characterization of IDPs [234]. A low-resolution protein structure defined through the ab initio modeling of SAXS data alone can be further refined using inputs from protein structure prediction tools, such as I-TASSER, CORAL, etc. [235,236]. Recently, various protein structure determination/prediction techniques and SAXS have been used to characterize partially disordered mycobacterial ESX-secretion-associated protein K (EspK) [237].

#### 3.2.5. Cryo-Electron Microscopy (Cryo-EM)

In the last five years, the research area involving the structural characterization of proteins and other biological entities has been revolutionized by the development of cryo-electron-microscopy-based techniques [238,239,240,241]. These methods overcome the limitations of primary methods, i.e., X-ray crystallography and NMR, and allow the structural characterization of relatively large, structurally heterogeneous, flexible, and dynamic assemblies at sub-nanometer atomic resolution (below 4 Å) [241,242,243]. Typically, a cryo-EM workflow contains three main steps: (a) vitrification (rapid cooling without ice crystal formation) of specimens in an aqueous solution, (b) image acquisition at a low electron dose using electron microscopy, and (c) 3D model reconstruction and validation. Single-particle analysis (SPA) and sub-tomogram averaging (STA) models are most commonly used for the structural annotation of proteins [244]. However, while the globular/ordered/structured regions of proteins can be structurally resolved using cryo-EM, the predicted intrinsically disordered regions in the proximity of flexible regions escape structural assignment [245]. Therefore, similar to x-ray crystallography, a high degree of intrinsic disorder restricts the implementation of cryo-EM techniques. Alternatively, the structure and dynamics of IDPs/IDRs can be investigated by complementing higher-resolution NMR studies of IDRs with the modeling capabilities of cryo-EM [246,247]. In conclusion, 3D cryo-EM maps in conjunction with high-resolution data from NMR can model IDPs under physiologically relevant conditions and provide insights into their functional behavior [243].

## 4. Computational Tools for Disorder Prediction

The biased amino acid compositions and peculiar sequence characteristics of IDPs/IDRs have encouraged the development of various reliable computational tools for studying intrinsic protein disorders. As a result, disorder predictors have been grouped into three distinct classes based on the underlying concepts.

### 4.1. Propensity-Based Predictors

In principle, a disorder predictor is classified as propensity-based if it depends on some essential physical or chemical characteristics of residues or on prior knowledge of the biological background of intrinsic protein disorder. Disorder-predicting tools, such as FoldIndex, NORSp, GlobPlot, CH plot, and PreLink belong to this category [45,245,246,247,248,249,250,251].

### 4.2. Machine Learning Algorithms (MLAs) Based Predictors

This class of advanced predictors relies on algorithms trained on data sets of experimentally characterized disordered regions and can differentiate disorder and order encoding sequences [29]. Currently, the experimentally characterized disordered proteins are publicly available on three databases: MobiDB (http://mobidb.bio.unipd.it/; accessed on 7 November 2022), IDEAL (https://ngdc.cncb.ac.cn/databasecommons/database/id/198; accessed on 7 November 2022), and DisProt (http://www.disprot.org/; accessed on 7 November 2022) [92,252,253]. PONDR, Spritz, DisEMBL, RONN, and DISOPRED are a few predictors that fit into this category [254,255,256,257,258,259].

Recently, the field of protein structure prediction has been revolutionized by the development of the deep learning-based method AlphaFold [260]. This software generates a per-residue confidence score (pLDDT) based on the protein’s amino acid sequence. The most recent version of this tool, i.e., Alphafold2, has been reported to achieve protein structure prediction accuracy competitive with that of experimental determination [261,262,263]. However, this program gives a low confidence score (pLDDT < 50) for intrinsically unstructured or disordered proteins/regions, and the inconclusive predicted structure resembles a ribbon. In addition, this method does not anticipate the relative likelihood of diverse IDP conformations and the folding pathways followed by IDPs/IDRs attaining an ordered structure upon interaction with other biomolecules [70,264]. At present, the AlphaFold Protein Structure Database is considered as the most complete and precise representation of the human proteome [265,266].

### 4.3. Inter-Residue Contact-Based Predictors

Predictors based on the idea that IDPs/IDRs are disordered because they cannot make enough inter-residue contacts required to compensate for the loss of configurational entropy during folding are grouped together as inter-residue contact-based predictors. The above conclusions may be derived by either simple statistics involving contact numbers or through sophisticated techniques of determining the total stabilization energy of a protein. Computational tools, such as IUPred, FoldUnfold, and Ucon belong to this class [267,268,269,270].

At present, there is no “best” disorder prediction computational tool. Therefore, to avoid the limitations of a given tool, prediction results from different disorder predictors relying on distinct principles should be combined to provide a consensus prediction, as implemented by meta-predictors (for example, PONDR-FIT) [271]. Alternatively, publicly available meta-servers (for example, MeDor and metaPRDOS can also be used for quick and simultaneous analysis of protein disorder using multiple predictors [272,273].

In several recent articles, extensive comparisons of various computational disorder prediction methods’ performance and comprehensive online resources useful for studying IDPs/IDRs were provided [274,275,276].

## 5. Evolution of IDPs/IDRs

The evolution of proteins involves changes in the form of insertions, deletions, or substitutions in their amino acid sequences. Over time, such changes can accumulate in the proteins, giving rise to taxonomic classes having substantial differences in their amino acid compositions [277]. In general, the structure and function of proteins are well conserved, but several exceptions exist. Several previous studies suggested that, even if the protein sequence diverges extensively, the protein function is well-conserved [278]. Hence, proteins are generally considered as the ‘chemical fingerprints’ of evolutionary history, as they manifest the underlying genetic changes as amino acid sequences.

The evolution of intrinsic disorder exhibits a wavy pattern in which highly disordered primordial proteins with predominantly RNA-chaperone-like activities were slowly replaced with highly structured proteins [118,279]. Later, because of its peculiar features regarding the regulation of complex cellular processes, protein disorder was reinvented at various succeeding evolutionary stages, resulting in the creation of more complex organisms from the last universal ancestor [280,281].

Several mechanisms, such as de novo generation, horizontal gene transfer, and lateral gene transfer, can give rise to genes that encode IDPs [57,282]. Approximately 14% of Pfam domains, predicted to be mostly disordered and shared by many protein families, appear to have originated from domain duplications and module exchange between genes [196]. The high frequency of occurrence of tandemly repeated sequences in IDPs/IDRs suggests that the expansion of internal repeat regions (microsatellite and minisatellite coding regions) is another possible way by which the IDPs encoding genes arose [283,284]. Looking at the exceptional functional variability conferred to IDPs/IDRs due to the genetic instability of repetitive elements, the mechanism of the extension of repeat elements appears as the frequent method of disorder spread during evolution and rapid genomic changes in adaptation [36,285,286]. Furthermore, these IDPs/IDRs can also act as hot spots for mutations, leading to the loss of different functional modalities and thus resulting in various types of diseases, including cancer [287,288,289]. Seera and Nagarajaram have recently shown that the disease-causing missense mutations within IDRs reduce the overall conformation heterogeneity of the IDRs as compared to their wild type counterparts, and the few ‘locked’ dominant conformations presumably limit their interaction with the cognate partners [290].

Recent studies have shown that disordered protein segments are encoded by GC-enriched gene regions, which, in turn, directly correspond to the disorderedness of the encoded proteins [291,292]. This GC enrichment is due to the prevalence of amino acids coded by GC-rich codons (G, A, R, and P) in the disordered regions of proteins [291]. At the residue level, a relatively higher rate of evolutionary changes in the disordered regions of proteins was observed compared with that in the ordered/globular domains, as there were no structural constraints to maintaining a 3-D structure [293]. However, in certain cases, structured domains and disordered regions of proteins have been observed to co-evolve at higher rates [294,295]. Despite these rapid changes, the biological functions of the structured domains and disordered regions are always conserved [296]. Hence, a deeper understanding of the conformation ensemble–function relationship will help to decipher the evolutionary trajectory of IDPs.

Based on the conservation of sequences coding for protein disorder, disordered residues have been classified as constrained (both sequence and disorder are conserved) or flexible (only protein disorder is conserved). Together, constrained and flexible disorder residues are known as conserved disorder. On the other hand, if neither disorder nor the residues encoding it are conserved, such a disorder class is known as non-conserved disorder [297]. This integrated structural and evolutionary approach has recently been used to define the determinants of the functional adaptability of the neutrophin family of proteins involved in neuronal development [298].

Considering that the disordered regions in proteins have a distinct amino acid composition and evolutionary rate as compared with that of ordered regions, the substitution frequencies of residues in the disordered regions must also be distinct from those found in ordered regions. Thus, identifying the evolutionary and functional features of IDPs/IDRs has become a computational challenge, as most of the sequence analysis tools and parameter optimization procedures are aimed at ordered/structured regions of proteins. Recently, methods evaluating disordered proteins’ molecular features and sequence composition in a position-specific manner have been developed. These advancements have allowed researchers to pursue alignment-based evolutionary studies on IDPs/IDRs without aligning the residues discretely [299,300,301].

## 6. IDPs/IDRs in Diseases

Like structured proteins, the expression, localization, and interactions of intrinsically disordered proteins (IDPs) are also highly coordinated and regulated. Multiple checkpoints at various stages of the expression of IDPs-specific genes (from transcript synthesis to protein degradation) ensure the availability of IDPs in appropriate quantities and for the desired duration, preventing any ectopic interactions [302]. Several studies have shown the role of IDPs/IDRs in different human disorders, including diabetes, cancer, amyloidosis, neurodegenerative, and cardiovascular diseases [303,304]. Some well-studied examples of IDPs associated with human disease are p53, Mdm2, PTEN, c-Myc, AF4, BRCA1, EWS, Bcl-2, c-Fos, HPV oncoproteins, etc. [303,305,306,307]. Moreover, the deposition of ⍺-synuclein, tau, and amyloid-β proteins leads to Alzheimer’s disease, the accumulation of ⍺-synuclein results in Parkinson’s disease, and aggregates of PrP^SC^ cause prion diseases. The expansion of CAG triplet repeats in disease genes, which introduces disorder, results in the family of polyQ diseases, such as Kennedy’s disease, Huntington’s disease, etc. [308,309,310,311,312,313,314].

In the last two decades, the role of IDPs in human diseases has been actively studied, giving rise to new mechanistic findings that have led to the formation of the D^2^ concept (‘Disorder to Disorders’) [303]. Several comprehensive reviews and thematic series articles have been published covering the significance of IDPs in diseases [315,316,317]. For instance, Coskuner and Uversky described various hypotheses proposed to explain the molecular mechanisms of the pathogenesis of Alzheimer’s and Parkinson’s diseases and suggested the need for the development of new techniques through the integration of quantum and statistical mechanics, thermodynamics, bioinformatics, and machine learning approaches, which, in turn, may lead to the development of new experimental approaches [318,319,320,321,322,323,324,325]. However, at present, there are several limitations and challenges associated with in silico studies of IDP-associated neurodegenerative disorders [326,327]. Another study found that an NADH-stabilized 26S proteasomal complex could degrade IDPs efficiently. Therefore, the accumulation of disease-causing disordered proteins, such as tau, c-Fos, p53, etc., can be prevented by the selective degradation of IDPs in an ATP-independent manner [328]. Moreover, the analysis of components of the ATP-dependent ubiquitin-proteasome degradation system (UPS) revealed the importance of the disorder content and MoRFs of the complex in neurodegenerative disorders and cancers [329]. However, identifying key mutations, PTM sites, and functional motifs in the disordered regions, exploring the evolutionary history of IDPs involved in diseases, understanding the cooperative functioning of ordered and disordered domains, and dissecting the IDPs’ interactome are some of the many active research areas involving IDPs/IDRs and diseases [288,330,331,332,333,334].

## 7. IDPs/IDRs as Drug Targets

With increasing evidence of their involvement in molecular functions complementing globular domains, essential biological processes, protein–nucleic acid interactions, protein–protein interactions, and diseases, IDRs/IDPs have emerged as one of the prime targets for drug discovery or repurposing [142,335,336,337,338,339]. However, IDP characteristics, such as a lack of a sTable 3D structure, very high flexibility, conformational ensembles, susceptibility to proteolytic cleavage, protein aggregation, etc., limit the application of the most-established experimental assays and computational methods that would otherwise work for ordered/globular proteins [340,341,342,343,344]. Therefore, IDP-specific drug screening/development is mainly a tradeoff between binding affinity/specificity and the alternation in the functioning of disordered proteins with other features, such as solubility, crowding, efflux, metabolism, etc., a potentially relevant role [345].

Broadly, disordered proteins/regions have been used in drug development procedures by targeting their conformational changes, interactions, and self-aggregating behavior [346]. For example, the inhibitor 10058F4 of Myc proto-oncogene protein (MYC) binds to MYC and prevents conformational disorder-to-order transition, which, in turn, blocks MYC-MAX complex-driven tumorigenesis [33,347,348,349,350]. Similarly, Methyl-CpG-binding domain protein 2 (MBD2) inhibitors restrict the folding of MBD2 upon binding to its partner p66α. This MBD2-p66α is known to regulate the Mi-2/NuRD chromatin remodeling complex involved in promoting metastasis in various cancer cells through epithelial–mesenchymal transition (EMT) [351,352]. In contrast to ordered proteins, the protein–protein interactions involving IDPs offer uneven, shorter, compact, and more mimicable surfaces for the tighter binding of small drug molecules [353,354,355]. In recent times, potential drug molecules have been designed to target either the disordered segment or the binding region of the interacting molecule. For instance, nutlins binding to Mdm2 prevent the interaction of Mdm2 with the disordered regions of p53, which activates the p53 pathway, leading to apoptosis, cell-cycle arrest, and the inhibition of the uncontrolled cell growth of human tumor xenografts [356]. Additionally, an FDA-approved compound, trifluoperazine dihydrochloride, was found to bind to a disordered region of multifunctional protein nuclear protein 1 (NUPR1) and arrest pancreatic ductal adenocarcinoma (PDAC) development [357]. Moreover, the disordered proteins from pathogens can also be targeted to interrupt their interaction with host proteins, which they utilize for their survival and pathogenesis [358]. In a recent review, Santofimia et al. comprehensively described targeting IDPs in various protein–protein and protein–nucleic acid interactions involved in cancer [359]. Furthermore, compounds, such as curcumin, rosmarinic acid, ferulic acid, and safranal, have also been reported to prevent the aggregation of α-synuclein protein by binding to monomers, thus inhibiting the polymerization of these proteins, which results in various neuronal malignancies [360,361]. In summary, deciphering the sequence–ensemble–function relationship of IDPs/IDRs and the development of efficient computational modeling approaches will help to unravel the enormous potential of disordered proteins as drug targets.

## 8. Conclusions

The spread and versatility of proteins’ functions carried by intrinsically disordered regions within them are phenomenal. The overview of the protein structure–function relationship presented in this review with a focus on various aspects of intrinsically disordered proteins/regions can be very helpful in understanding the fundamentals of biologically active structureless proteins. It also offers a novel perspective for characterizing proteins with unknown functions.

Over the past two decades, the research field of protein disorder has witnessed a revolution with respect to the understanding of various aspects of IDPs/IDRs such as sequence, abundance, structure, function, regulation, evolution, etc. However, there is still a requirement for new theoretical, experimental, and computational models specific to intrinsically disordered regions in proteins that can explain both the diversifying behavior of IDPs and the unifying principle of protein structure–function relationships.

## Figures and Tables

**Figure 1 ijms-23-14050-f001:**
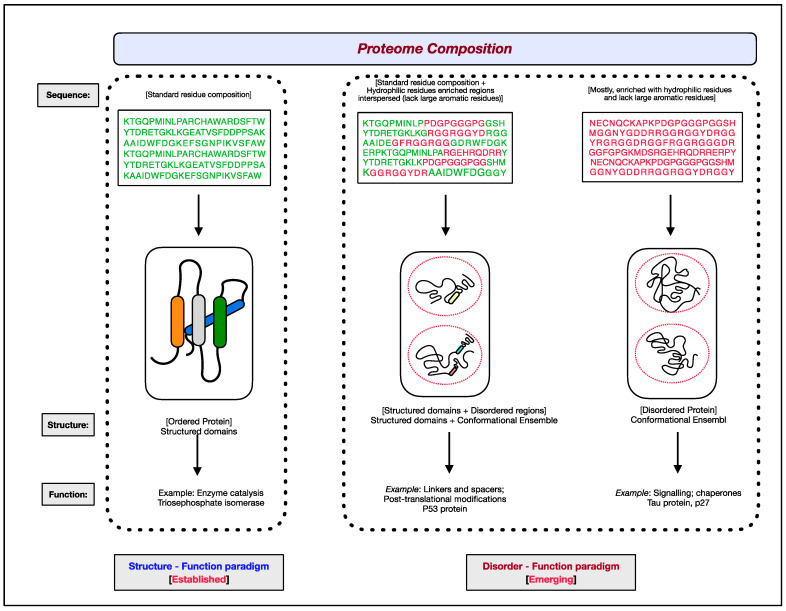
The two paradigms of protein structure and function. According to the well-established ‘structure–function paradigm’, a three-dimensional native structure under physiological conditions is vital for a protein to perform its biological function (for example, enzyme-catalyzed reactions). The more recent ‘disorder–function paradigm’ states that a protein can carry out its biological function without attaining a 3-D stable folded structure under physiological conditions (for example, protein binding to other cellular molecules). For representative purpose, residues coding for ordered/globular domains are shown in ‘green’ color, and residues coding for disordered proteins/segments are shown in ‘red’. At the proteome level, the structured domains and intrinsically disordered regions (IDRs) are two functional building blocks of proteins.

**Figure 2 ijms-23-14050-f002:**
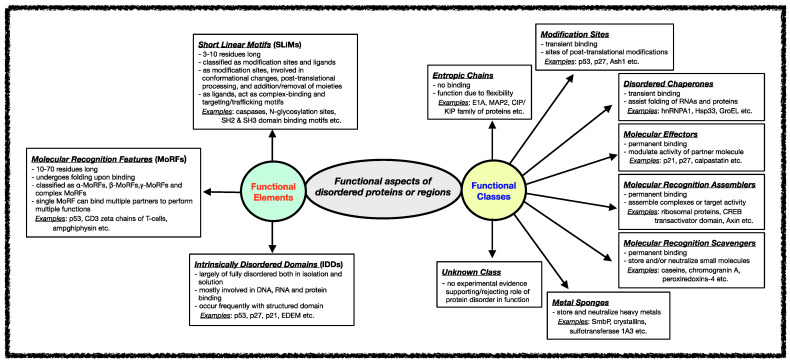
Functional aspects of IDPs/IDRs. Intrinsically disordered proteins/regions’ functional classes and elements are described here. The functional class scheme describes eight different categories into which IDPs/IDRs can be grouped based on their biological function. The different functional classes include Entropic chains, Modification sites, Disordered chaperones, Molecular effectors, Molecular recognition assemblers, Molecular recognition scavengers, Metal sponges, and Unknown. IDPs/IDRs’ functions are mediated mainly through three types of structural elements, namely Short Linear Motifs (SLiMs), Molecular Recognition Features (MORFs), and Intrinsically Disordered Domains (IDDS).

## Data Availability

Not applicable.

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
