# Peer review of "Intrinsically Disordered Proteins: An Overview"

_ijms, 2022, doi:10.3390/ijms232214050_

Round 1
Reviewer 1 Report
The authors have written a review with the title: ”Intrinsically Disordered Proteins: An Overview.” Intrinsically disordered proteins (IDP) or proteins containing intrinsically disordered domains (IDD) are substantial parts of many proteomes. They carry out numerous vital functions in living organisms. Until recently, the information about this class of proteins had been very sparse, mostly due to limited experimental tools for their study. Fortunately, during the last ten to fifteen years a progress in experimental techniques of structural biology have opened the access to the study of their physical-chemical properties and stimulated the interest in IDPs.
From this perspective, writing of such manuscript is the right time achievement. The authors have collected huge amount of data about IDPs (IDDs). The manuscript is well written, the text is logically ordered into chapters characterizing the IDPs from many aspects starting with the change of structure - function paradigm, continuing with their sequence and structural characteristics, functions and experimental techniques for their studying including computational tools for the prediction of disordered regions. Everything is well supported by numerous references. The manuscript is easily readable, the text is without any grammar and syntax errors. I have found only one, i.e. on line 521 where the word micosatellites should be corrected to microsatellites.
I have only one comment to the chapter 4.2 Machine Learning Algorithms. There is a well-done list of available computational tools for the predicting of disordered regions. I suggest adding Alpha Fold concept to this chapter. I know that the software is not primarily determined for the prediction of disordered regions, however, I believe that it deserves to be mentioned.
In conclusion, I fully support publishing the manuscript almost as it is, i.e. with minimal amendments.
Author Response
We thank the reviewer for the critical assessment of our manuscript. These valuable suggestions are indeed very useful, and guided us immensely to improve the quality of the manuscript. Responses to the reviewer’s comments are included below and highlighted in bold.
Comment 1: "I have found only one, i.e. on line 521 where the word micosatellites should be corrected to microsatellites."
Response: The word has been corrected in the marked sentence on line 521. The modified sentence in the revised manuscript appear as written below.
................................................................................................
"Higher frequency of occurrence of tandemly repeated sequences in IDPs/IDRs suggests that the expansion of internal repeat regions (microsatellites and minisatellites coding regions) as another possible way in which the IDPs encoding genes may have arisen (Tompa 2003; Lise and Jones 2005)." ................................................................................................ Comment 2: "I have only one comment to the chapter 4.2 Machine Learning Algorithms. There is a well-done list of available computational tools for the predicting of disordered regions. I suggest adding Alpha Fold concept to this chapter. I know that the software is not primarily determined for the prediction of disordered regions, however, I believe that it deserves to be mentioned." Response: We have added a paragraph about Alphafold in the section "4.2 Machine learning algorithms based predictors". Relevant references has been cited at the appropriate places. The modified section 4.2 in the revised manuscript appear as shown below. ................................................................................................
4.2. Machine Learning Algorithms (MLAs) Based Predictors
This class of advanced predictors relies on algorithms trained on data sets of experimentally characterized disordered regions, and capable of differentiating disorder and order encoding sequences (Tompa 2002; Habchi et al.2014). Currently the experimentally characterized disordered proteins are publicly available on three databases: MobiDB (http://mobidb.bio.unipd.it/) (Di Domenico et al. 2012); IDEAL (http://www.ideal.force.cs.is.nagoya-u.ac.jp/IDEAL/) (Fukuchi et al. 2012) and DisProt (http://www.disprot.org/) (Sickmeier et al. 2007). PONDR (Romero et al. 1999), Spritz (Vullo et al. 2006), DisEMBL (Linding et al. 2003a), RONN (Yang et al. 2005), DISOPRED (Ward et al. 2004) are few predictors that fits in this category.
Recently, the field of protein structure prediction has been revolutionized with the development of deep learning-based method AlphaFold (Jumper et al. 2021). This software generates a per-residue confidence score (pLDDT) based on the amino acid sequence of the given protein. The most recent version of this tool, i.e., Alphafold2 has been reported to achieve protein structure prediction accuracy competitive to that of experimental determination (Zweckstetter et al. 2021; AlQuraishi et al. 2021). However, this program gives a low confidence score (pLDDT < 50) for intrinsically unstructured or disordered proteins/regions, and the inconclusive predicted structure looks-like ribbon. In addition, this method does not anticipate the relative likelihood of diverse IDPs conformations, and the folding pathways followed by IDPs/IDRs attaining an ordered structure upon interaction with other biomolecules (Ruff et al. 2021; Strodel et al. 2021). At present, the AlphaFold Protein Structure Database has been considered as the most complete and precise representation of human proteome (Varadi et al. 2022; Tunyasuvunakool et al. 2021). ................................................................................................
Reviewer 2 Report
Title: Intrinsically disordered proteins: An overview
This review describes intrinsically disordered proteins, their classifications, functions, experimental methods and roles in disease. This is a large and interesting field that has gained a lot of momentum in the last 20 years. Unfortunately, this review provides very little new information, as is evident by the fact that only about 15 references out of more than 200 were published after 2014. This date is significant because an excellent, comprehensive review on the subject was published in 2014 in Chem. Rev. 114, 6589-631. The authors borrow very heavily from this review, essentially paraphrasing entire sections of it. This is particularly evident in sections 2.4 and 2.5 where the order and titles of subheadings are identical to those in the earlier review (sections 2 to 3.3), and the same examples are often presented. The two figures are also virtually identical to those in Chem. Rev. While the authors apparently got permission to use them, I do not see the benefit in publishing them twice. Overall, this review presents very little new information relative to previous reviews and in a generally less comprehensive and detailed presentation. Perhaps if the authors chose to focus on a subset of newer data (e.g. intrinsic proteins in disease), it would be more successful.
Author Response
We thank the reviewer for the critical assessment of our manuscript. These valuable suggestions are indeed very useful, and guided us immensely to improve the quality of the manuscript. Responses to the reviewer’s comments are as follows:
Point 1: This review describes intrinsically disordered proteins, their classifications, functions, experimental methods and roles in disease. This is a large and interesting field that has gained a lot of momentum in the last 20 years. Unfortunately, this review provides very little new information, as is evident by the fact that only about 15 references out of more than 200 were published after 2014. This date is significant because an excellent, comprehensive review on the subject was published in 2014 in Chem. Rev. 114, 6589-631. The authors borrow very heavily from this review, essentially paraphrasing entire sections of it. This is particularly evident in sections 2.4 and 2.5 where the order and titles of subheadings are identical to those in the earlier review (sections 2 to 3.3), and the same examples are often presented.
Response 1: As per the suggestion of the reviewer, we have reframed the review article (both titles and content), and incorporated concepts and examples based on the recent literature (published after 2014). The functional aspects of IDPs/IDRs (sections 2.4 and 2.5) have been extensively modified, and discussed in more stratified manner. In the present format, approximately 50% of references have been published after 2017. This review has been written with a purpose to benefit broad spectrum of researchers by providing both fundamental generalizations and recent findings in the field of disordered proteins. After incorporating reviewer’s suggestions, we firmly believe that the revised manuscript has been improvised tremendously and meet the purpose.
Point 2: The two figures are also virtually identical to those in Chem. Rev. While the authors apparently got permission to use them, I do not see the benefit in publishing them twice.
Response 2: As per the suggestion of the reviewer, we created figures in the revised manuscript. While Figure 1 summarizes the two paradigms of protein structure and function from sequence, structure and function perspectives, Figure 2 represents different functional classes and elements of intrinsically disordered proteins/regions. Both the figures have been attached as a single pdf file and also in revised manuscript.
Point 3: Overall, this review presents very little new information relative to previous reviews and in a generally less comprehensive and detailed presentation. Perhaps if the authors chose to focus on a subset of newer data (e.g., intrinsic proteins in disease), it would be more successful.
Response 3: As per the suggestion of the reviewer, we have incorporated new concepts, explanations and examples in the manuscript. As a result, the revised manuscript is the amalgam of both fundamental and advanced knowledge about intrinsically disordered proteins/regions. We have also made extensive incorporations and modifications in the chapters 6 and 7 explaining the role of IDPs/IDRs in diseases and its utility as a potential drug target respectively. Therefore, we firmly believe that the manuscript in the present format presents recent findings in the comprehensive manner.
Point 4: Moderate English changes required.
Response 4:
As per the suggestion of the reviewer, english modifications have been made in the entire manuscript for easy readability and understanding purpose

Reviewer 3 Report
The manuscript entitled "Intrinsically Disordered Proteins: An Overview” is an interesting review and worth publishing in IJMS. I think the paper has a good scientific sound and can be useful for biologists, specifically structural biologists.
The authors also gave an overview of the approaches used to study “structurally” the IDPs and/or IDRs, but it is mandatory to add a session in the “Experimental techniques” concerning the direct one. Nowadays, small-angle X-ray scattering (SAXS) is an important technique for dealing with IDPs and IDRs. SAXS can be used to estimate if the protein is flexible and disordered or partially disordered but can also do some modelling in the cases using the information from experimental techniques, mentioned by authors, crystallography, and NMR but also from some structure prediction software (f.es iTASSER). I suggest adding also some indication about cryo-EM. They can get a look into this paper “Gijsbers et al Structural Analysis of the Partially Disordered Protein EspK from Mycobacterium tuberculosis, Crystals 2021, 11, 18. https://doi.org/10.3390/cryst11010018
Author Response
We thank the reviewer for the critical assessment of our manuscript. These valuable suggestions are indeed very useful, and guided us immensely to improve the quality of the manuscript. Response to the reviewer’s comments has been included below and highlighted in bold.
Comment: The authors also gave an overview of the approaches used to study “structurally” the IDPs and/or IDRs, but it is mandatory to add a session in the “Experimental techniques” concerning the direct one. Nowadays, small-angle X-ray scattering (SAXS) is an important technique for dealing with IDPs and IDRs. SAXS can be used to estimate if the protein is flexible and disordered or partially disordered but can also do some modelling in the cases using the information from experimental techniques, mentioned by authors, crystallography, and NMR but also from some structure prediction software (f.es iTASSER). I suggest adding also some indication about cryo-EM. They can get a look into this paper “Gijsbers et al Structural Analysis of the Partially Disordered Protein EspK from Mycobacterium tuberculosis, Crystals 2021, 11, 18. https://doi.org/10.3390/cryst11010018
Response: As per the suggestion of the reviewer, in the section "3 Experimental Approaches for Assessing Intrinsic Protein Disorder", we have included subsections 3.2.4 and 3.2.5 describing "Small-Angle X-Ray Scattering (SAXS)" and "Cryo-Electron Microscopy" respectively. We have also included the relevant references at appropriate places. The included subsections are as follows:
....................................................................................................................
3.2.4 Small-Angle X-Ray Scattering (SAXS)
In solution, SAXS technique can quickly define structural characteristics of proteins of sizes ranging from few kilodaltons to several gigadaltons under various experimental setups (Kikhney et al. 2015, Korasick et al. 2018, Da vela et al. 2020, Grawert et al. 2020). Briefly, this method involves exposing the sample placed in quartz capillary tubes to a collimated monochromatic X-ray beam source and capturing of scattered photons by the detector (Mertens et al. 2010, Jacques et al. 2010). Comparative analysis of the electron density distributions of the protein sample and pure solvent/buffer is then used to determine various parameters of proteins in solution such as molecular mass, volume, radius of gyration, folding state etc (Kikhney et al. 2015). Moreover, SAXS data can also be used to define protein flexibility and intrinsically disordered state of proteins in solution (Grawert et al. 2020, Czaplewski et al. 2021). Most commonly, the scattering profiles of proteins obtained from SAXS experiments, and represented as Kratky plots [s2I(s) as a function of s, where s and I represent momentum transfer function and scattering intensity] is used to get structural insights about the protein. In contrast to globular proteins bell-shaped Kratky’s plot with well-defined maxima, disordered proteins-specific Kratky plots appear as plateau for a given range of momentum transfer function (s), followed by a monotonic increase (Perez et al. 2001, Bernado et al. 2009). Additionally, IDPs experimentally determined radius of gyration (Rg) from SAXS curve can be directly compared with a globular and random coil theoretical or experimental Rg values for a given number of residues. The Rg values of IDPs lies between highly compact globular proteins (lowest Rg values) and completely disordered/unfolded proteins represented by random coils (highest Rg values) (Bernado et al. 2012). Altogether, this method offers fast structural characterization of proteins in solutions with relatively easy sample preparation protocol, and capture of data under near-native conditions (Jeffries et al. 2015, Graewert et al. 2017). Since the sensitivity of SAXS depends on the particle size, prior removal of macromolecules aggregates during sample preparation using methods like sedimentation, size-exclusion chromatography is suggested (David et al. 2009).
Finally, it's worth mentioning that SAXS-based studies of IDPs can use a priori valuable complementary information from several other experimental and in silico methods of protein structure determination. For instance, X-ray crystallography depicts structured regions of protein, while SAXS defines the protein segments with missing electron density (Schulte et al. 2014). Similarly, during analyses of bimolecular complexes and multi-domain proteins, NMR provides information about different domains/complex sub-units, and SAXS defines relative inter-domain positions (Rodriguez et al. 2020). Furthermore, other complementary techniques such as CD, spectroscopy, chromatography etc., along with SAXS can be used for biophysical characterization of IDPs (Gast et al. 1995). Low-resolution protein structure defined through ab initio modeling of SAXS data alone can be further refined using inputs from protein structure prediction tools such as I-TASSER, CORAL etc. (Zhang et al. 2008, Petoukhov et al. 2012). Recently, various protein structure determination/prediction techniques along with SAXS has been used to characterize partially disordered mycobacterial ESX-secretion associated protein K (EspK) (Gijsbers et al. 2021).
3.2.5 Cryo-Electron Microscopy (Cryo-EM)
In the last 5 years, research arena involving structural characterization of proteins and other biological entities has been revolutionized by the development of cryo-electron microscopy-based techniques (Nogales et al. 2016, Shoemaker et al. 2018, Hanske et al. 2018, Nwanochie et al. 2019). These methods overcome the limitations of primary methods i.e., x-ray crystallography and NMR, and allows structural characterization of relatively large, structurally heterogenous, flexible, and dynamic assemblies at sub-nanometer atomic resolution (below 4 Å) (Cheng et al. 2018, Nwanochie et al. 2019, Musselman et al. 2021). Typically, a cryo-EM workflow contains three main steps: (a) vitrification (rapid cooling without ice crystals formation) of specimens in aqueous solution, (b) image acquisition at low electron dose using electron microscopy, and (c) 3D model reconstruction and validation. Most commonly, single particle analysis (SPA) and sub-tomogram averaging (STA) models are used for structural annotation of proteins (Bharat et al. 2015). While the globular/ordered/structured regions of proteins are structurally resolved using cryo-EM, the predicted intrinsically disordered regions in the proximity of flexible regions escapes structural assignment (Yan et al. 2015). Therefore, similar to x-ray crystallography, the presence of high degree of intrinsic disorder restricts the implementation of cryo-EM techniques. Alternatively, IDPs/IDRs structure and dynamics can be investigated by complementing higher resolution NMR studies of IDRs with modeling capabilities of cryo-EM (Gibbs et al. 2017, Geraets et al. 2020). In conclusion, 3D cryo-EM maps in conjunction with high resolution data from NMR has the capability to model IDPs in physiologically relevant conditions and provide insights about their functional behavior (Musselman et al. 2021).
.....................................................................................................................
Round 2
Reviewer 2 Report
The authors have rewritten large portions of the manuscript and added many more modern examples. It is a significant improvement over the previous version. I only have access to the track-changes manuscript, and there may be an issue with how the figures are presented. I see the old figure along with a truncated version of the new figure. Provided this is only a track changes issue, what I can see of the new figures looks fine. There are still a few typos/grammatical errors (e.g. "confirmation" rather than "conformation" on line 353), so the manuscript could use a final proofread.
Author Response
Point-by-point response to the reviewer’s comment
We thank the reviewer for the critical assessment of our manuscript. The suggestions are indeed very useful, and guided us immensely to improve the quality of our manuscript. Response to the reviewer’s comments has been addressed below, and changes are marked up in the revised manuscript using ‘track changes’ (Font color: Red).Response to Reviewer 2 comments:
Point 1: The authors have rewritten large portions of the manuscript and added many more modern examples. It is a significant improvement over the previous version. I only have access to the track-changes manuscript, and there may be an issue with how the figures are presented. I see the old figure along with a truncated version of the new figure. Provided this is only a track changes issue, what I can see of the new figures looks fine.
Response 1: The alignment of figures appears to be a track-change issue in the revised manuscript pdf file generated. The figures are positioned appropriately in the revised manuscript word file submitted. For your reference, we have enclosed the Figure1 and Figure2.
Point 2: There are still a few typos/grammatical errors (e.g., "confirmation" rather than "conformation" on line 353), so the manuscript could use a final proofread.
Response 2: As per the suggestion of the reviewer, we performed a final proofread of the manuscript. The typos and grammatical errors at various places have been corrected in the revised version of the manuscript.
